# Proteolytic Cleavage of Bioactive Peptides and Protease-Activated Receptors in Acute and Post-Colitis

**DOI:** 10.3390/ijms221910711

**Published:** 2021-10-02

**Authors:** Michelle De bruyn, Hannah Ceuleers, Nikita Hanning, Maya Berg, Joris G. De Man, Paco Hulpiau, Cedric Hermans, Ulf-Håkan Stenman, Hannu Koistinen, Anne-Marie Lambeir, Benedicte Y. De Winter, Ingrid De Meester

**Affiliations:** 1Laboratory of Medical Biochemistry, University of Antwerp, 2610 Wilrijk, Belgium; michelle.debruyn2@uantwerpen.be (M.D.b.); anne-marie.lambeir@uantwerpen.be (A.-M.L.); 2Infla-Med, Centre of Excellence, University of Antwerp, 2610 Wilrijk, Belgium; hannah.ceuleers@uantwerpen.be (H.C.); nikita.hanning@uantwerpen.be (N.H.); maya.berg@uantwerpen.be (M.B.); joris.deman@uantwerpen.be (J.G.D.M.); benedicte.dewinter@uantwerpen.be (B.Y.D.W.); 3Laboratory of Experimental Medicine and Pediatrics (LEMP), University of Antwerp, 2610 Wilrijk, Belgium; 4Bioinformatics Knowledge Center (BiKC), Howest University of Applied Sciences, 8000 Bruges, Belgium; Paco.Hulpiau@howest.be (P.H.); hermans.cedric@howest.be (C.H.); 5Department of Clinical Chemistry and Haematology, University of Helsinki and Helsinki University Hospital, 00290 Helsinki, Finland; ulf-hakan.stenman@helsinki.fi (U.-H.S.); hannu.k.koistinen@helsinki.fi (H.K.); 6Department of Gastroenterology and Hepatology, Antwerp University Hospital (UZA), 2650 Edegem, Belgium

**Keywords:** protease activity, PAR, thrombin, trypsin, β-endorphin, enkephalins, neurotensin, inflammation, IBD, IBS

## Abstract

The protease activity in inflammatory bowel disease (IBD) and irritable bowel syndrome has been studied extensively using synthetic fluorogenic substrates targeting specific sets of proteases. We explored activities in colonic tissue from a 2,4,6-trinitrobenzenesulfonic acid (TNBS)-induced colitis rat model by investigating the cleavage of bioactive peptides. Pure trypsin- and elastase-like proteases on the one hand and colonic tissue from rats with TNBS-induced colitis in the acute or post-inflammatory phase on the other, were incubated with relevant peptides to identify their cleavage pattern by mass spectrometry. An increased cleavage of several peptides was observed in the colon from acute colitis rats. The tethered ligand (TL) sequences of peptides mimicking the N-terminus of protease-activated receptors (PAR) 1 and 4 were significantly unmasked by acute colitis samples and these cleavages were positively correlated with thrombin activity. Increased cleavage of β-endorphin and disarming of the TL-sequence of the PAR3-based peptide were observed in acute colitis and linked to chymotrypsin-like activity. Increased processing of the enkephalins points to the involvement of proteases with specificities different from trypsin- or chymotrypsin-like enzymes. In conclusion, our results suggest thrombin, chymotrypsin-like proteases and a set of proteases with different specificities as potential therapeutic targets in IBD.

## 1. Introduction

To date, the underlying pathophysiology of irritable bowel syndrome (IBS) and inflammatory bowel disease (IBD) is not entirely elucidated, but a key role for proteases has been described [1,2,3,4,5,6,7,8,9,10,11,12,13]. The involvement of these proteases in IBS and IBD has been studied by evaluating their expression and activity in rodent models and patients.

Colonic tissues from IBS and IBD patients have shown an increased trypsin-like activity measured with Boc-QAR-AMC [4,8]. Likewise, it was shown that colonic samples from animal models of dextran sulfate sodium (DSS) and 2,4,6-trinitrobenzenesulfonic acid (TNBS)-induced colitis in the acute phase, as a model for IBD, and in colonic samples during the post-inflammatory (post-colitis) phase, as a model for IBS, showed increased trypsin-like activity as well. This activity was measured with small fluorogenic and chromogenic substrates [10,12,13].

A first trypsin-like protease to be involved in IBD is thrombin. An increased thrombin activity was measured in colonic samples from patients with Crohn’s disease. This increased activity and an increased thrombin expression were also observed in colonic tissue from TNBS-induced acute colitis rats [11]. A second trypsin-like protease reported to be involved in both IBD and IBS is tryptase [2,3,4,14]. In colonic samples from IBD patients, an increased number of tryptase-positive mast cells was detected [14]. Furthermore, colonic biopsies from IBS patients showed an elevated tryptase concentration and an increased tryptase release from mast cells [2,3,4]. A final trypsin-like protease with increased mRNA expression in the colon of IBS patients is trypsin itself (isoform not specified) [4]. Further research showed that the expression of trypsin-3 was upregulated in colonic tissue of IBS patients [8].

In addition to an increased trypsin-like activity, an increased elastase-like activity was measured in human colonic tissue and DSS or TNBS-induced colitis mice [10,15]. In situ zymography on colonic biopsies from IBD patients showed an increased elastase activity compared to colon from healthy individuals. Elastase-2 is probably responsible for this increased activity since the level of this protein is also increased in colonic IBD tissues. Similarly, increased mRNA expression and an increased immunostaining for elastase-2 were observed in colonic tissue from DSS-induced colitis mice [15].

In fecal samples from patients with IBS and IBD, an increased total protease activity was measured using an azocasein assay [5,6,7,16]. Furthermore, fecal supernatant from IBS patients showed increased trypsin-like, chymotrypsin-like, neutrophil elastase, proteinase-3 and cathepsin G activity [9,16]. Fecal samples from a TNBS-induced post-colitis rat model also showed an increased elastase-like activity [13].

The studies cited above evidently emphasize the importance of trypsin-like and elastase-like proteases in feces and colonic tissue obtained in the acute and post-inflammatory phase of colitis. Therefore, we first aimed to obtain cleavage patterns of naturally occurring substrates with a set of purified trypsin-like and elastase-like proteases present in the human colon (trypsin-1, -2, -3, tryptase, thrombin, cathepsin G, neutrophil elastase and pancreatic elastase). We focused on β-endorphin, vasoactive intestinal peptide (VIP), neurotensin, enkephalins, substance P and bradykinin because these peptides are linked to pain sensitization or inflammation and, as such, are relevant in the pathophysiology of colitis [17,18,19,20,21,22,23,24,25,26,27,28,29,30]. Since a role for protease-activated receptors (PARs) in colitis and post-colitis has been described, the ability of the proteases to ‘unmask’ or ‘disarm’ PARs was assessed as well [31,32,33,34]. In vivo, the proteolytic cleavage of PARs to reveal the tethered ligand (TL) sequence as new N-terminus is called ‘TL-sequence unmasking’ and leads to receptor activation. If the PARs are cleaved within the TL-sequence itself or if this sequence is removed, the TL-sequence is ‘disarmed’, and the receptor is inactivated [32,35].

As a second aim, we explored proteolytic activities in colonic tissues in an unbiased way by investigating the processing of the above-mentioned bioactive peptides and PAR N-termini. The peptide cleavages by colonic tissue are compared between control rats and rats in the acute or post-inflammatory phase of TNBS-induced colitis. With respect to substrates, such an unbiased approach is novel as previous studies have utilized fluorogenic or colorimetric substrates addressing the activity of a single protease or a set of proteases with similar specificity. We explored correlations between our peptide-based findings and activities measured with fluorogenic substrates. In this way, we want to contribute to the unraveling of the pathophysiology of IBD and IBS and to unravel the identity of specific proteases involved that deserve to be further explored as potential therapeutic targets.

## 2. Results

### 2.1. Proteolytic Cleavage of Bioactive and PAR-Based Peptides by Purified Enzymes

Proteolytic cleavage patterns of selected peptides involved in pain or inflammation were detected by mass spectrometry after incubation with cathepsin G and a set of purified trypsin-like and elastase-like enzymes. The ability of the proteases to unmask or disarm PAR N-termini was assessed as well. Here, synthetic peptides based on the N-terminal amino acid sequence of the four known PARs were used. These PAR-derived peptides contain the N-terminal sequence up to ten amino acids beyond the known TL-cleavage site. A summary of the efficiency of the included proteases to cleave the different peptides in vitro is shown in Table 1.

#### 2.1.1. β-endorphin

β-endorphin was cleaved very rapidly by trypsin-2. After 2 min of incubation, fragments could be detected resulting from the following cleavage sites: YGGFMTSEK↓SQTPLVTLFK↓NAIIK↓NAYKKGE (Figure 1a). Incubation with trypsin-1, tryptase and cathepsin G resulted in the same three cleavages, but a fourth cleavage site, after lysine in the 28th position, was introduced as well. The first cleavage in the processing of β-endorphin occurred after 2 min for all three proteases. Other cleavage sites were observed after 5 and 10 min for trypsin-1, after 5 min for tryptase and after 5 and 30 min for cathepsin G. Furthermore, the dual specificity (lysine and phenylalanine) of cathepsin G was illustrated with three different chymotrypsin-like cleavage sites in addition to the trypsin-like cleavage sites above. Thrombin and trypsin-3 did not cleave β-endorphin even after 30 min of incubation.

Neutrophil and pancreatic elastase both cleaved β-endorphin after threonine in position 16 already after 2 min of incubation. Neutrophil elastase also cleaved after valine in position 15. Cleavage sites in β-endorphin for each protease are shown in Figure 1a.

#### 2.1.2. Vasoactive Intestinal Peptide

As shown in Figure 1b, incubation of VIP with trypsin-1, trypsin-2 and tryptase resulted in three trypsin-like cleavage sites: HSDAVFTDNYTR↓LR↓K↓QMAVKKYLNSILN. This processing occurred more rapidly with trypsin-1 and -2 (after 2 min of incubation) than with tryptase, where the first cleavage was observed after 2 min, but the other cleavages could only be observed after 10 min. The cleavage after arginine in position 12 was also observed after 2 min of incubation with the same concentration of trypsin-3, while the cleavage after arginine in position 14 was observed after only 30 min. The dual specificity of cathepsin G was highlighted also with this peptide since one trypsin-like and one chymotrypsin-like cleavage was observed. Interestingly, the trypsin-like cleavage with cathepsin G differed from the cleavage sites of trypsins and tryptase. Thrombin did not cleave VIP, even after 30 min of incubation.

Pancreatic and neutrophil elastase cleaved VIP after threonine in the seventh position, after 2 and 10 min of incubation, respectively: HSDAVFT↓DNYTRLRKQMAVKKYLNSILN. Neutrophil elastase also showed a cleavage site after valine in position 19 already after 2 min of incubation. Cleavage sites in VIP for each protease are depicted in Figure 1b.

#### 2.1.3. Substance P, Bradykinin, Neurotensin, Leu-enkephalin and Met-enkephalin

No fragments of substance P, bradykinin or neurotensin were detected after incubation with the trypsin-like enzymes. However, a decrease in the relative intensity of the full-length peptide (compared to standard peptide) was observed after 10 min of incubation for substance P, bradykinin and neurotensin with all the trypsin-like enzymes and cathepsin G. Thus, considering the low molecular weight of these peptides and the limitation of the MALDI-TOF/TOF instrument to only detect peptides larger than 500 Da, it is possible that fragments are too small for detection. Neutrophil and pancreatic elastase were not incubated with these peptides since cleavage sites for both proteases are absent (Table 1 N/A). The relative intensity of both Leu- and Met-enkephalin was only decreased after 30 min of incubation with pancreatic elastase. Trypsin-like proteases were not incubated with the enkephalins since trypsin-like cleavage sites (arginine and lysine) are absent in the peptides (Table 1 N/A).

#### 2.1.4. PAR1-Based Peptide

Tryptase cleaved the PAR1-based peptide at four different cleavage sites: ARTR↓AR↓RPESK↓ATNATLDPR↓SFLLRNPNDK (Figure 2a). The cleavage after lysine already occurred immediately after the addition of tryptase to the peptide, while the other cleavages were observed after 2 min of incubation. The same cleavages, except for the cleavage after arginine in position 20, were observed after 2 min of incubation with trypsin-1 and -2 as well. Thrombin also processed the PAR1-based peptide very fast but cleaved only after the arginine in position 20. Trypsin-3 and cathepsin G cleaved the peptide at two sites, after lysine 11 and arginine 20. With trypsin-3, the cleavage after lysine was observed after 2 min of incubation and after arginine after 5 min. Opposite to this, cathepsin G cleavage after arginine was observed after 2 min of incubation and then at 5 min, after lysine. The elastase-like proteases cleaved the PAR1-based peptide at two different cleavage sites after threonine and alanine: ARTRARRPESKAT↓NA↓TLDPRSFLLRNPNDK. Both cleavages were observed after 10 min of incubation with pancreatic elastase. Neutrophil elastase cleaved the peptide at the same alanine and threonine after 5 and 10 min, respectively. The cleavage sites of all proteases are shown in Figure 2a.

All trypsin-like enzymes included in this experiment cleaved the PAR1-based peptide after arginine in position 20 and thus unmasked the TL-sequence. Thrombin did this immediately, followed by tryptase, trypsin-1, trypsin-2 and cathepsin G. Trypsin-3 revealed the TL-sequence more slowly. The cleavages with neutrophil or pancreatic elastase were rather slow and did not unmask or disarm the TL-sequence of the PAR1-based peptide.

#### 2.1.5. PAR2-Based Peptide

Trypsin-1, trypsin-2 and tryptase cleaved the PAR2-based peptide after arginine at position 12, unmasking the TL-sequence, already after 2 min of incubation. Trypsin-1 and trypsin-2 also showed two other cleavage sites resulting in TIQGTNR↓SSK↓GR↓SLIGKVDGTSHV (Figure 2b). The cleavage after arginine in position 7 was observed with tryptase as well. Neutrophil elastase processed the peptide only after 30 min of incubation by cleavage after valine in position 18, resulting in the disarming of the TL-sequence. Trypsin-3, thrombin, cathepsin G and pancreatic elastase did not cleave the peptide after 30 min of incubation. Cleavage sites of the PAR2-based peptide after incubation with the enzymes are highlighted in Figure 2b.

#### 2.1.6. PAR3-Based Peptide

The PAR3-based peptide was cleaved after lysine in position 17 to unmask its TL-sequence by trypsin-1, trypsin-2 and thrombin after already 2 min of incubation: GMENDTNNLAKPTLPIK↓TFRGAPPNSFE. Furthermore, except for thrombin, all the proteases also cleaved the peptide at cleavage sites that resulted in disarming of the receptor. The inactivation resulted from cleavage after threonine 18 by neutrophil and pancreatic elastase, cleavage after phenylalanine in position 19 by cathepsin G or cleavage after arginine in the 20th position by all three trypsins and tryptase: GMENDTNNLAKPTLPIKT↓F↓R↓GAPPNSFE. Cleavage sites of the enzymes in the PAR3-based peptide are shown in Figure 2c.

#### 2.1.7. PAR 4-Based Peptide

The cleavage unmasking the TL-sequence of the PAR4-based peptide was observed after 2 min of incubation with trypsins and thrombin: GGTQTPSVYDESGSTGGGDDSTPSILPAPR↓GYPGQVCAND. The abundance of the N-terminal fragment resulting from this cleavage was highest with thrombin after 30 min of incubation. Tryptase, cathepsin G, neutrophil and pancreatic elastase did not process the PAR4-based peptide after 30 min of incubation. These results are shown in Figure 2d.

### 2.2. Proteolytic Cleavage of Bioactive and PAR-Based Peptides by Colonic Samples from Acute and Post-Colitis Models

Next, we determined cleavage patterns of the same substrates after incubation with colonic tissue lysates from a TNBS-induced colitis rat model (n = 8). Overall, the cleavage of a set of these peptides was different with the samples from the acute or post-inflammatory phase of colitis and points to proteases as potential therapeutic targets.

#### 2.2.1. Decreased Processing of Neurotensin by Acute and Post-Colitis Samples

There was a significant decrease of the full-length neurotensin only after 20 min of incubation with acute colitis samples (*p* = 0.02), indicating a rather slow process. No significant decrease of the full-length peptide was observed in control or post-colitis samples, although a trend was seen in the post-colitis group (*p* = 0.06) (Figure 3b). However, the abundance of the full-length peptide was already significantly lower in control samples after seconds of incubation (at the starting point) compared to acute (*p* = 0.003) and post-colitis samples (*p* = 0.05) (Figure 3a). This might indicate very rapid processing of neurotensin in control samples. Peptide fragments were not detected, suggesting that neurotensin is cleaved in fragments too small to be detected with the MALDI technique.

#### 2.2.2. Increased Processing of Leu-enkephalin and β-endorphin by Acute Colitis but Not Post-Colitis Samples

After 5 min of incubation, the abundance of the full-length Leu-enkephalin was significantly lower in acute colitis samples compared to control (*p* = 0.05) or post-colitis samples (*p* = 0.02) (Figure 4b). However, degradation of the peptide was only significant after 10 min of incubation with control (*p* = 0.04) and post-colitis samples (*p* = 0.05), but not with acute colitis samples (Figure 4c). Therefore, we hypothesize that Leu-enkephalin is degraded very fast, masking further cleavage in acute colitis samples. In support of this hypothesis, there was a trend towards decreased full-length peptide at the starting point in acute colitis samples compared to control (*p* = 0.1) or post-colitis (*p* = 0.06) (Figure 4a). Leu-enkephalin contains two chymotrypsin-like cleavage sites (tyrosine and phenylalanine), but there was no correlation between the abundance of the full-length Leu-enkephalin and the chymotrypsin-like (*p* = 0.4) or cathepsin G activity (*p* = 0.3) measured using fluorogenic substrates.

β-endorphin was cleaved more intensively by proteases in acute colitis samples compared to control and post-colitis samples. Two cleavage sites were observed: one after leucine 17 and one after phenylalanine 18 (Figure 5a).

The first cleavage site, at leucine 17, resulted in C- and N-terminal fragments that were both more abundant in acute colitis samples compared to controls (*p* = 0.02 for the N-terminal fragment and *p* = 0.001 for the C-terminal fragment) after 10 min of incubation. The C-terminal fragment was also significantly elevated in acute colitis compared to post-colitis samples (*p* = 0.001) after 10 min. Furthermore, after 30 min of incubation the C- and N-terminal fragments in acute colitis samples were more abundant compared to both control (*p* = 0.02 and *p* = 0.004) and post-colitis samples (*p* = 0.005 and *p* = 0.04). These results are shown in Figure 5b1,b2.

The second cleavage, after phenylalanine 18, resulted in C-and N-terminal fragments that were not detected in control samples even after 30 min of incubation. Since other fragments pointing towards a cleavage after this phenylalanine were not observed either, it can be concluded that this cleavage did not occur in colonic samples from control animals. Moreover, the formed N-terminal fragment was increased in acute colitis samples compared to post-colitis samples after 10 (*p* = 0.02) and 30 (*p* = 0.001) min of incubation and the C-terminal fragment was elevated in acute colitis samples compared to post-colitis samples after 30 min (*p* = 0.02). Results are shown in Figure 5c1,c2.

The two observed cleavage sites in β-endorphin are chymotrypsin-like and a positive correlation was observed between chymotrypsin-like activity measured with a fluorogenic substrate and the abundancy of the formed N- and C-terminal fragments after cleavage at leucine 17 (R = 0.0551; *p* = 0.005 for the N-terminal fragment and R = 0.452; *p* = 0.03 for the C-terminal fragment). For the cleavage after phenylalanine 18, the correlation between the chymotrypsin-like activity and the abundance of the N-terminal fragment was significant (R = 0.530; *p* = 0.008), but the correlation with the C-terminal fragment was not (*p* = 0.1).

Results from Section 2.1.1. showed cleavage after phenylalanine 18 by cathepsin G. Therefore, it can be hypothesized that cathepsin G might be responsible for the increased β-endorphin cleavage after phenylalanine 18 in acute colitis samples. However, there were no significant correlations between cathepsin G activity and the abundance of the N- and C-terminal fragment after cleavage after F18 (*p* = 0.4 and *p* = 0.9).

#### 2.2.3. Increased Processing of Met-enkephalin and Bradykinin by Acute Colitis and Post-Colitis Samples

Both Met-enkephalin and bradykinin were significantly processed by proteases in acute and post-colitis samples, but not in control samples. No peptide fragments of Met-enkephalin could be detected because of the limitation of the MALDI technique to detect short peptides. However, the abundance of full-length peptide was significantly reduced after 10 min of incubation with acute (*p* = 0.01) or post-colitis samples (*p* = 0.02) but not with control samples (*p* = 0.1) (Figure 6). The sequence of Met-enkephalin contains two chymotrypsin-like cleavage sites, but there was no correlation between the decrease of the full-length peptide and the chymotrypsin-like activity measured with fluorogenic substrates (*p* = 0.3). The results for bradykinin can be found in the Appendix A.

#### 2.2.4. Similar Processing of VIP and Substance P by Control, Post-Colitis and Acute Colitis Samples

VIP was cleaved by the proteases in the colonic samples after leucine 13 and arginine 14: HSDAVFTDNYTRL↓R↓KQMAVKKYLNSILN. There were no differences in VIP processing between the groups after 10 and 30 min of incubation except for an increased abundancy of the N-terminal fragment resulting from cleavage after arginine 14 in acute colitis samples compared to controls after 10 min of incubation (*p* = 0.05) (Appendix A). Substance P proved to be relatively stable in colonic samples. A decreased abundance was only observed after 30 min of incubation (Appendix A). There were no differences between controls, acute and post-colitis samples, and no peptide fragments could be detected.

#### 2.2.5. Activation’ or ‘Disarming’ of PAR-Based Peptides

The PAR1-based peptide was cleaved after arginine in position 20 to unmask the TL-sequence: ARTRARRPESKATNATLDPR↓SFLLRNPNDK. After 30 min of incubation, this occurred with six out of eight acute colitis and three out of eight post-colitis samples, but not with control samples. The increase of the fragment representing this TL-sequence was significant in acute colitis samples (*p* = 0.03), but only a trend towards increase could be observed in post-colitis samples (*p* = 0.1) (Figure 7a). The abundance of this fragment after 30 min was positively correlated with the trypsin-like activity measured with Tos-GPR-AMC (R = 0.438; *p* = 0.03). A positive correlation of the abundancy of this fragment after 10 (R = 0.722; *p* = 0.00007) and 30 min (R = 0.688; *p* = 0.0002) of incubation was also observed with the thrombin activity in all colonic tissues (n = 24).

Incubation of the PAR2-based peptide with colonic samples resulted in two different cleavage sites: TIQGTNRSSKG↓R↓SLIGKVDGTSHV. Cleavage after the arginine in position 12 unmasks the TL-sequence. There were no differences in the abundance of the peptide representing this sequence (Figure 7b). Hence, proteases in colonic samples from all three groups can activate the receptor to the same extent. However, this cleavage was not observed in all samples. Proteases from two out of eight control, three out of eight post-colitis and four out of eight acute colitis samples could unmask the TL-sequence from the PAR2-based peptide. As mentioned in Section 2.1.5, trypsin-1, -2 and tryptase can be responsible for this PAR2-activating cleavage. The cleavage after glycine in position 11 was not observed with the purified enzymes.

The PAR3-based peptide was cleaved slowly by proteases in the colonic samples after phenylalanine in the 19th position, thereby disarming the TL-sequence: GMENDTNNLAKPTLPIKTF↓RGAPPNSFE. The increase of the N- and C-terminal fragment was significant after incubation with acute colitis samples (*p* = 0.03 for both). There was a trend towards an increase of the C-terminal fragment (*p* = 0.07) but not the N-terminal fragment (*p* = 0.4) after incubation with post-colitis samples. No C- or N-terminal fragments could be detected after incubation with control samples, except for one. These results are shown in Figure 7(c1,c2).

This disarming PAR3-cleavage was mentioned earlier (Section 2.1.6.) after incubation with cathepsin G. However, there was no significant correlation between the abundance of the fragments after cleavage and the cathepsin G activity measured with a fluorogenic substrate. Interestingly, there was a correlation between the abundance of the N-terminal fragment and the chymotrypsin-like activity measured with suc-AAPF-AMC (R = 0.407; *p* = 0.05). It can thus be hypothesized that chymotrypsin-like enzymes, different from cathepsin G, are responsible for the disarming cleavage in the PAR3-based peptide.

The PAR4-based peptide was cleaved after arginine 30 to unmask the TL-sequence: GGTQTPSVYDESGSTGGGDDSTPSILPAPR↓GYPGQVCAND. The N-terminal fragment was observed after incubation with two out of eight post-colitis and five out of eight acute colitis samples but not in control samples. The increase of this fragment was significant in acute colitis samples (*p* = 0.04) but not in post-colitis samples (*p* = 0.2) (Figure 7d). The C-terminal fragment was not detected. Incubation with trypsins and thrombin resulted in this cleavage site as well (paragraph 2.1.7.). Furthermore, there was a significant correlation between the trypsin-like activity measured with Tos-GPR-AMC and the abundance of the N-terminal fragment after 10 min of incubation (R = 0.406; *p* = 0.05). The abundance of the same fragment after 10 min (R = 0.574; *p* = 0.003) and 30 min (R = 0.656; *p* = 0.0005) of incubation was also positively correlated with the thrombin activity in all colonic tissues (n = 24).

## 3. Discussion

There is ample evidence in literature for the involvement of proteases in IBS and IBD [1,2,3,4,5,6,7,8,9,10,11,12,13]. Most studies rely on the use of synthetic substrates for trypsin-like and elastase-like enzymes that are often not capable of making further refinements towards the specific proteases involved [4,8,9,10,11,12,13,16]. We used the cleavage patterns of peptides based on in vivo occurring substrates to further identify specific proteases pointing to a set of proteases as potential targets for therapy in the acute and post-inflammatory phase of a TNBS-induced colitis rat model.

### 3.1. Proteolytic Cleavage of Bioactive and PAR-Based Peptides by Purified Enzymes

First, we used pure proteases to determine the cleavage patterns of the peptides that were based on in vivo substrates. Different trypsin-like and elastase-like proteases, which have been proposed to be involved in IBS and IBD [2,3,4,8,9,10,12,13,14,15,16], cleaved the peptides with different efficiencies. Both VIP and β-endorphin could be used to differentiate between the studied proteases. The dual specificity that distinguishes cathepsin G from the other trypsin-like proteases had already been described before and recent literature confirmed this by using chromogenic peptide substrates and substrate phage display [36,37,38,39]. After incubation of β-endorphin with pure proteases, we observed four trypsin-like cleavages that were reported in literature after incubation of the peptide with trypsin for 60 min [40]. Our experiments showed differences in the processing of this peptide by trypsin-1, trypsin-2 and trypsin-3. To the best of our knowledge, this is the first report on the processing of β-endorphin by different trypsin isoforms, tryptase and cathepsin G.

Schilling and coworkers previously used peptide-based specificity profiling to distinguish between trypsin-1/2 and trypsin-3. Trypsin-1 slightly favored lysine over arginine in the P1 position, while trypsin-3 did not discriminate between these amino acids [41]. However, we observed no cleavage by trypsin-3 after lysine residues that were susceptible for cleavage by trypsin-1 and -2 in all the investigated peptides, except for the PAR1-based peptide. Furthermore, only slow processing of VIP after arginine in position 14 was observed by trypsin-3 under our experimental conditions, compared to ultrafast processing by trypsin-1 and trypsin-2. This difference between trypsin-1 or -2 and trypsin-3 could not be explained by the previously reported specificity profiling of the human trypsins [41]. The differential cleavage of these peptides can be used to understand which trypsin-like enzymes are active in a given biological sample. Since both VIP and β-endorphin contain several trypsin-like cleavage sites (arginine or lysine), it was surprising to observe no cleavage of VIP and β-endorphin by thrombin.

Regarding the elastase-like proteases, it is known that pancreatic elastase I and III preferably cleave substrates after alanine, valine, isoleucine or threonine [42,43,44,45]. Pancreatic elastase II, also named epithelial elastase, prefers cleavage after leucine, methionine, phenylalanine or tyrosine [43,46]. Neutrophil elastase is known to cleave after valine, isoleucine, threonine and alanine [47,48,49]. In our experiments, an elastase preparation containing pancreatic elastase I from porcine pancreas and a preparation with human neutrophil elastase were used. As expected, our results show cleavage after alanine, isoleucine and threonine by both pancreatic and neutrophil elastase. However, cleavage after valine was only seen after incubation with neutrophil elastase. In contrast with the literature [42,43,44,45,46,47,48,49], our results suggest that the preference of neutrophil elastase to cleave after valine in VIP and β-endorphin can distinguish between pancreatic and neutrophil elastase.

Regarding the peptides based on the PARs, we showed that thrombin, tryptase, trypsin-1, trypsin-2, trypsin-3 and cathepsin G could unmask the TL-sequence of the PAR1-based peptide. This occurred more rapidly by thrombin compared to the other proteases. Trypsin-1, trypsin-2 and tryptase unmasked the TL-sequence of the PAR2-based peptide. We also expected but did not observe this unmasking after incubation with trypsin-3 [8]. The physiological concentration of trypsin-3 used in our experiment (25 nM) is probably too low to induce this cleavage. Although it has been described that only 10 nM of trypsin-3 could signal to human neurons through a PAR2-dependent mechanism, 360 nM of trypsin-3 was necessary to process a PAR2-based peptide [8,50].

Similarly, our results did not show unmasking of the PAR4-based peptide by cathepsin G. This is in contrast with published data describing increased permeability in mice due to PAR4 activation by cathepsin G after intracolonic administration of fecal supernatant from patients with ulcerative colitis (UC) [51]. However, it can be hypothesized that the cathepsin G concentration used in our experiments is lower than in fecal supernatant and too low to unmask the TL-sequence. The findings of Sambrano et al. contribute to this hypothesis since 500 nM of cathepsin G was needed to activate PAR4 [52]. The physiological relevance of this high cathepsin G concentration can be questioned since we measured low cathepsin G activity in colon from control and colitis animals (Appendix A). We did observe unmasking of the TL-sequence of the PAR4-based peptide by trypsin-1, trypsin-2, trypsin-3 and thrombin. We also showed that this cleavage was most abundant after incubation with thrombin.

### 3.2. Proteolytic Cleavage of Bioactive and PAR-Based Peptides by Colonic Samples from Acute and Post-Colitis Models

Next to pure proteases, we determined cleavage patterns of in vivo occurring substrates after incubation with colonic tissue lysates from a TNBS-induced colitis rat model. Cleavage of a set of these peptides was different in the acute or post-inflammatory phase of colitis and pointed to proteases as potential therapeutic targets. Cleavage of substance P, VIP and the PAR2-based peptide was not different in the acute or post-colitis animals compared to controls. The processing of the enkephalins and β-endorphin was increased in acute colitis samples, while the processing of neurotensin occurred very rapidly in control samples. Furthermore, the unmasking of the TL-sequence of the PAR1- and PAR4-based peptides and the disarming of the TL-sequence of the PAR3-based peptide was significant in the colonic tissue from acute colitis rats as well. The cleavage of Met-enkephalin and bradykinin was also significant after incubation with colon from post-colitis animals. A summary is shown in Figure 8.

#### 3.2.1. Cleavage of Substance P, VIP and the PAR2-Based Peptide Is Not Different in the Acute or Post-Inflammatory Phase of TNBS-Induced Colitis

The processing of Substance P, VIP and the unmasking of the TL-sequence of the PAR2-based peptide were not different between control, acute or post-colitis animals. It has been shown that substance P is involved in gastrointestinal inflammation and mucosal injury, but we did not observe significant differences in the processing of this peptide between colonic samples from control, acute colitis and post-colitis animals [28].

The role of VIP has been studied in IBD and IBS, but conflicting results have been observed. In IBD, reports of an increased VIP level in colon from patients and enhanced communication between mast cells and VIP contradict other literature describing a decrease or no changes in the level of VIP in colon from individuals with IBD [53,54,55,56,57]. In IBS, it has been shown that the colonic level of VIP is increased in biopsies from female IBS-D patients [58]. Other authors did not observe differences regarding VIP levels in colonic lysate [19].

We did not observe differences in the degradation of VIP by proteases in colonic samples from controls, acute colitis and post-colitis animals. Although, we did expect an increased peptide processing at the trypsin-like cleavage site (arginine in position 14) by samples from post-colitis animals since we previously demonstrated an increased trypsin-like activity measured with fluorogenic substrates [12,13]. We hypothesize that the proteases responsible for VIP degradation are different from the proteases leading to an increased trypsin-like activity measured by fluorogenic substrates. We showed a very rapid VIP processing with trypsin-1, trypsin-2 and tryptase. If these proteases led to the increased trypsin-like activity in colon from post-colitis animals measured with fluorogenic substrates [12,13], we probably would have observed increased VIP processing by these samples as well. We also showed no processing of VIP by thrombin and only very slow processing at arginine 14 by trypsin-3. Therefore, thrombin and trypsin-3 could be responsible for an increased trypsin-like activity in colon from post-colitis animals measured with fluorogenic substrates, while no increased VIP degradation was observed. This is in line with literature, reporting trypsin-3 as one of the proteases responsible for increased trypsin-like activity in patients with IBS [8]. An increased thrombin expression has been described in colonic tissue from IBS-D patients [59]. In this pathophysiological condition, thrombin can lead to visceral hypersensitivity symptoms by activating PAR1 [60]. As only a limited set of the known trypsin-like proteases has been included in our experiments with purified enzymes, we cannot exclude the importance of other trypsin-like proteases.

It has been shown that PAR2 activation can result in increased visceral hypersensitivity and anti- or pro-inflammatory effects [34,61]. Trypsin-3 can induce visceral hypersensitivity by activation of PAR2 and it is proposed that this protease is involved in the pathophysiology of IBS [8]. Therefore, we did expect to see a difference in PAR2 processing between post-colitis animals and acute colitis or control animals [31,32,33,34]. However, no differences were observed between the control, post-colitis and acute colitis samples regarding the processing of the PAR2-based peptide. All colonic samples were able to reveal the TL-sequence as the new N-terminus to a similar extent.

Discrepancies between our results and literature regarding peptide processing or PAR activation in vivo can be due to the limitations of our research. Here, peptides were added in excess and conditions such as co-factor concentration or pH may differ from the in vivo situation. Further, the proteases and peptides used here are based on the sequence of human peptides to facilitate further translational research, but the peptides are processed by proteases from rat colonic tissue. In rats, four forms of trypsin have been observed, including anionic trypsin-1, anionic trypsin-2, cationic trypsin-3 and trypsin-4. However, the amino acid sequence of these proteases is 70% to 80% identical to the sequences from human trypsin-1 and -2, and a protease similar to human trypsin-3, p23, has been reported in rats [62,63,64,65,66]. Both trypsin-3 and p23 are resistant to soybean trypsin inhibitor (SBTI) and can cleave PAR1 and PAR2 to promote inflammation and pain [50]. The sequences of rat and human thrombin are 87% identical [67,68].

#### 3.2.2. Cleavage of β-endorphin and Disarming of the TL-Sequence of the PAR3-Based Peptide in Acute Colitis Point to the Involvement of Chymotrypsin-Like Proteases

Endogenous opioid peptides, such as β-endorphin and enkephalins, have known antinociceptive, anti-diarrheal and anti-inflammatory effects [69,70]. A decreased formation or an increased degradation of these endogenous peptides can attenuate their effects. Our experiments showed a significantly increased processing of β-endorphin by proteases in colonic samples from acute colitis animals. The link between increased processing of this peptide and colonic inflammation is in line with the faster processing of β-endorphin after incubation with inflamed rat paw tissue [40]. Literature also describes decreased levels of this peptide in colonic tissue from mice with acute colitis compared to chronic DSS-induced colitis [71]. Furthermore, colonic biopsies from patients with active UC showed significantly lower levels of β-endorphin compared to patients with chronic UC [72].

Regarding the proteases involved in β-endorphin degradation by colonic tissue, we did not observe trypsin-like cleavage sites. We did show a positive correlation between the presence of β-endorphin degradation products and chymotrypsin-like activity measured with suc-AAPF-AMC. These findings suggest the involvement of proteases with a chymotrypsin-like specificity in acute colitis. According to the MEROPS database, the protease insulysin can process β-endorphin into β-endorphin 1–17 or β-endorphin 1–18 [73]. This intracellular protease can be released after cell death during colitis or after in vitro cell lysis during our experiments [74]. A second protease, cathepsin D, is known to cleave β-endorphin at leucine in position 17 [75]. Indeed, an increased expression of cathepsin D in colonic biopsies from IBD patients compared to controls was previously observed [76]. Furthermore, the inhibition of cathepsin D ameliorated inflammatory scores in the acute DSS colitis mouse model [77]. Further research is necessary to explore if this increased cathepsin D expression in acute colitis affects inflammation and pain by the degradation of β-endorphin.

Next to the cleavage of β-endorphin, the disarming of the TL-sequence of the PAR3-based peptide was significant in acute colitis samples but not in control or post-colitis samples. The abundance of the fragment resulting from this cleavage was also positively correlated with the chymotrypsin-like activity measured with suc-AAPF-AMC. Literature reported an increased chymotrypsin-like activity in fecal samples from IBD patients and rats with TNBS-induced colitis [9,78]. Cathepsin G was also labeled as a hyperactive protease in colonic tissue from IBD patients [79]. However, we did not observe a correlation between cathepsin G activity and the processing of β-endorphin or the PAR3-based peptide. It is important to note that the disarming cleavage of the PAR3-based peptide in our experimental setting is no proof for the inactivation of PAR3 in a physiological setting.

#### 3.2.3. Unmasking of the TL-Sequence of the PAR1- and PAR4-Based Peptide in Acute Colitis Points to the Involvement of Thrombin

PARs have been described in relation to visceral hypersensitivity and gut inflammation [31,32,33,34]. PAR1 and PAR4 are upregulated in colonic biopsies from IBD patients and activation of these receptors is linked to pro-inflammatory effects [51,58]. Our results showed significantly increased unmasking of the PAR1- and PAR4-based peptides by proteases in colon from acute colitis. Unexpectedly, this trypsin-like cleavage was not found to a similar extent in post-colitis samples [12,13]. The absence of an increased trypsin-like cleavage in post-colitis samples is similar to the results with VIP and in both cases, it is probable that the enzymes responsible for an increased trypsin-like activity with fluorogenic substrates differ from the proteases that process PAR1, PAR4 and VIP.

Our results showed that the TL-unmasking of the PAR1- and PAR4-based peptides observed in acute colitis samples occurred more efficiently with thrombin compared to the other trypsin-like proteases. Thrombin unmasked the TL-sequence of the PAR1-based peptide already after seconds of incubation, where the other trypsin-like proteases needed minutes of incubation. In addition, the fragment resulting from the TL unmasking of the PAR4-based peptide was more abundant after incubation with thrombin than the other proteases. Furthermore, the TL-unmasking of both peptides by colonic tissue was positively correlated with the trypsin-like activity measured with Tos-GPR-AMC but not Boc-QAR-AMC. The Michaelis–Menten constant (Km) for both these substrates and trypsin-1, trypsin-2, trypsin-3 and thrombin were compared (Appendix A). The Km for thrombin with Boc-QAR-AMC was considerably higher compared to the trypsins, indicating that the affinity of thrombin for Boc-QAR-AMC is lower compared to trypsin-1, trypsin-2 and trypsin-3. Furthermore, the TL-unmasking of PAR1 and PAR4 was positively correlated with the specific thrombin activity measured using Tos-GPR-AMC and dabigatran. Taken together, these findings point to thrombin as a potential target in the acute phase of colitis and strongly corroborate recent literature describing an increased thrombin activity in colonic samples from patients with Crohn’s disease and rats with TNBS-induced colitis. Furthermore, thrombin caused mucosal damage in mice through mechanisms that involved PAR1 and PAR4 and was mentioned as a hyperactive protease in the colonic mucosa of IBD patients [11,79]. A role for PAR4 in colitis was also suggested in a T-cell transfer colitis mouse model, where treating the mice with the serine protease inhibitor UAMC-00050 ameliorated intestinal inflammation and permeability and decreased the mRNA expression of PAR4 [80]. Although our results suggest the involvement of thrombin in acute colitis, our experimental set-up does not allow us to draw conclusions on altered PAR1 or PAR4 activation in vivo. In the first instance, our experimental set-up aimed to reveal proteolytic activities able to cleave peptides relevant in IBD and IBS among which are the four PAR-based peptides. To reveal the involvement of PAR activation in IBD or IBS, functional activation assays are needed using for instance PAR-transfected cells [1,4,5,51,60,81].

#### 3.2.4. Cleavage of Met-enkephalin, Leu-enkephalin and Neurotensin Point to the Involvement of Proteases with Other Specificities

We hypothesized ultrafast processing of Leu-enkephalin by proteases in colon from acute colitis animals and we observed an increased processing of Met-enkephalin in acute colitis and post-colitis samples. The increased degradation of these endogenous opioid peptides can diminish their effects in animals with acute or post-colitis. It is known that these peptides have a short lifespan because of the degradation by neprilysin and aminopeptidase N [69]. Inhibitors of these proteases might increase the local level of the peptides and prolong their effects. They are thus attractive targets for therapy in acute and post-colitis [22,69]. The enkephalins contain chymotrypsin-like cleavage sites but no correlation between chymotrypsin-like or cathepsin G activity and the peptide degradation was observed. Therefore, next to trypsin- and chymotrypsin-like enzymes, a role for proteases with different specificities is probable. Dipeptidyl peptidase I, neprilysin-2 and carboxypeptidases A4 and A6 are known to cleave one of or both the enkephalins [82,83,84,85].

The role of neurotensin in IBD has been mainly studied by the administration of exogenous neurotensin. This neuropeptide exerts a dual effect depending on the time of administration. In animal studies, administration of neurotensin before colitis induction results in the promotion of colitis development. Treatment with neurotensin after colitis induction reduced mucosal damage, suggesting a role for neurotensin in mucosal healing following colitis [86,87,88]. Our results suggest very rapid cleavage of neurotensin by proteases in control samples but only slow cleavage by proteases in acute and post-colitis samples. The in vivo relevance of this interesting finding should be investigated in further studies.

In conclusion, we first incubated a set of peptides selected based on their in vivo relevance in colonic inflammation and pain, with different trypsin-like and elastase-like proteases. This resulted in different cleavage patterns and efficiencies, even between trypsin isoforms. Next, we described the cleavages of bioactive peptides by colonic tissue from a TNBS-induced colitis model in the acute and post-inflammatory phase, pointing to a set of proteases as potential targets for therapy. First, the increased processing of β-endorphin and disarming of the PAR3-based peptide in acute colitis samples were correlated to the chymotrypsin-like activity, suggesting the involvement of chymotrypsin-like proteases. Second, the increased unmasking of the TL-sequence of the PAR1- and PAR4-based peptide in acute colitis was linked to the specific thrombin activity and therefore points to a role for thrombin. Finally, alongside trypsin- and chymotrypsin-like enzymes, a role for proteases with a different specificity is suggested by the processing of the enkephalins. Further research regarding these proteases and their natural substrates can contribute to the unraveling of the role of specific proteases in IBD and IBS and their potential as therapeutic targets.

## 4. Materials and Methods

### 4.1. Animal Model

Adult male Sprague–Dawley rats (200–225 g; Charles River, Wilmington, MA, USA) were used in a previously described TNBS-induced acute and post-inflammatory model [12,13,89,90]. Rats were housed at constant room temperature (22 ± 2 °C) and humidity (60%) with two rats per cage, unlimited access to water and food and on a 12 h–12 h light/dark cycle. All experiments were approved by the Ethical Committee for Animal Experiments of the University of Antwerp (EC nr. 2014-41 and nr. 2018-79).

After an overnight fast, TNBS colitis was induced using a TNBS-enema (4 mg TNBS; Sigma-Aldrich, 50% ethanol) as described previously [12,13,89,90]. Rats were anesthetized with a mixture of ketamine (35 mg/kg i.p.; Ketalar^®^, Pfizer, New York City, NY, USA) and xylazine (5 mg/kg i.p.; Rompun^®^, Bayer, Leverkusen, Germany), and the TNBS-enema or 0.9% saline was administered intrarectally using a flexible catheter (18 G, length 4.5 cm) to colitis and control rats, respectively. Animals were kept in a tail-up position for 1 min and were then allowed to recover in their cages with free access to water and food. Three days after induction of colitis, acute colitis was confirmed by colonoscopy, rats from the control and acute colitis group (n = 8) were sacrificed and colonic samples were taken for further experiments. For the post-colitis group (n = 8), a colonoscopy was performed starting from day 10 and repeated every 4 days until complete mucosal healing was observed. The rats were then sacrificed as well and colonic samples were taken.

### 4.2. Preparation of Colonic Lysate

Colonic lysates were prepared for studying their effects on cleavage patterns of added peptides with a MALDI-TOF/TOF assay. After sacrifice, a sample of approximately 50 mg of distal colon was taken from each rat. The colonic samples were immediately rinsed with Krebs solution, snap-frozen and stored at −80 °C until further processing. Sample preparation started with crushing the colonic samples on dry ice. Liquid nitrogen was used to prevent the loss of enzymatic activity due to temperature increases and the colon powders were stored at −80 °C for further experiments. Next, the frozen colon powders were dissolved in lysis buffer (1% octyl glucoside; Sigma-Aldrich, 120 mM NaCl in 50 mM Tris-HCl pH 8.0) for 15 min and centrifuged (12,000× *g*) for 5 min at 4 °C. After centrifugation, the supernatants were collected and used immediately to prepare the reaction mixtures, measure proteolytic activities with fluorogenic substrates and determine the protein content with a Bradford assay.

### 4.3. Proteolytic Activity with Fluorogenic Substrates

Trypsin-like, chymotrypsin-like, elastase-like, kallikrein-like, cathepsin G and thrombin activity were measured in colonic samples from the TNBS-induced colitis model using fluorogenic substrates (Appendix A). Except for thrombin activity, activities measured in colon from the TNBS-induced post-colitis model have been published elsewhere by Ceuleers et al. and Hanning et al. [12,13].

Trypsin-like activity was determined using Boc-QAR-AMC (75 µM) and Tos-GPR-AMC (100 µM) in 50 mM Tris-HCl and 120 mM NaCl pH 8.0. Elastase-like activity was measured using both suc-AAPV-AMC (450 µM) and suc-AAA-AMC (450 µM) and chymotrypsin-like and kallikrein activities were measured with suc-AAPF-AMC (450 µM) and H-PFR-AMC (450 µM) respectively. All substrates were purchased from Bachem (Bubendorf, Switzerland). Cathepsin G activity was determined using suc-AAPF-AMC and cathepsin G inhibitor I (1 µM; Calbiochem, San Diego, CA, USA) by making the difference in activity without and with inhibitor. Thrombin activity was measured using Tos-GPR-AMC and dabigatran (1 µM; Sigma, Saint Louis, MO, USA) in the same manner in 50 mM Tris-HCl and 120 mM NaCl pH 8.3.

Preheated substrate was added to colonic lysate to start the reaction. Fluorescence was measured for 20 min at 37 °C on a Tecan Infinite F200 pro microtiter plate reader. A calibration curve with AMC (Sigma-Aldrich, Saint Louis, MO, USA) was used to determine activities in units per liter, where a unit is defined as the amount of enzyme that catalyzes the conversion of one micromole of the substrate per minute in the specific experimental conditions described. Activities were normalized to protein concentration determined with Bradford to obtain activities in units per gram protein. Km for trypsin-1, trypsin-2, trypsin-3 and thrombin with Boc-QAR-AMC and Tos-GPR-AMC were also determined (Appendix A).

### 4.4. Proteolytic Cleavage of Natural Substrates

#### 4.4.1. Peptides and Proteases of Interest

Peptides of interest were based on the sequences of the four known PARs and synthesized by Pepscan. Other peptides involved in pain and inflammation were also included and purchased from Bachem. Purity of all peptides was above 95% and the peptide sequences can be found in Table 2.

Trypsin-like proteases included in the experiments are trypsin-1, -2 and -3, tryptase (Enzo LifeSciences), thrombin (Sigma-Aldrich) and cathepsin G (Sigma-Aldrich). Trypsin-1, -2 and -3 were produced as described by Koistinen et al. and Wu et al. [91,92]. Two elastase-like enzymes, neutrophil elastase (Enzo LifeSciences, Farmingdale, NY, USA) and pancreatic elastase I (Sigma-Aldrich) were included as well. All enzymes were from human origin except for pancreatic elastase from porcine pancreas.

#### 4.4.2. Reaction Mixtures

Colonic lysate was added to a peptide of interest to obtain the reaction mixtures with the colonic samples. Lysate and peptide were diluted so each mixture contained 10 µM peptide of interest and 20 µg/mL protein originating from the colonic tissue. These peptide and protein concentrations were based on research from Loew et al. and O’ Donoghue et al. [93,94]. The mixtures were incubated at 37 °C for several incubation periods between 5 and 30 min. Afterward, the fragmentation reaction was stopped by adding trifluoroacetic acid (TFA; Alfa Aesar™, ThermoFisher, Waltham, MA, USA) to a final concentration of 2% and standard peptide (substance P, bradykinin or PAR3-based peptide) was added in a final concentration of 10 µM. Standard peptides were selected based on the mass of the peptides of interest. For each sample/peptide combination, a starting point was included as well. Here, TFA was added immediately after mixing the reaction constituents. Samples were stored at −80 °C for further MALDI-TOF/TOF experiments.

Reaction mixtures with purified enzymes were compiled in an identical way. Each mixture contained an equimolar concentration of protease (25 nM based on [41,50,93,94]) and 10 µM peptide of interest and was incubated for 2, 5, 10 and 30 min at 37 °C. Substance P was added as the standard peptide except for the reaction mixtures with substance P or bradykinin as the peptide of interest, where angiotensin II (Bachem) was added.

#### 4.4.3. MALDI-TOF/TOF

The TFA mixtures were thawed on ice and desalted using C18 zip-tips (Merck Millipore^®^, Burlington, MA, USA) with a capacity of 5 µg protein. Retained peptide fragments were eluted directly on the MALDI target plate. Next, 0.5 µL of a 2.5 mg/mL α-cyano-4-hydroxycinnamic acid (Sigma-Aldrich) matrix solution (in 70% acetonitrile, 0.1% TFA) was spotted on top of the sample. A 6-Peptide mixture Kit (AB Sciex, Framingham, MA, USA) containing bradykinin (2–9), angiotensin I, glu1-fibrinopeptide, ACTH (1–17), ACTH (18–39) and ACTH (7–38) was diluted with matrix and spotted on the plate as well.

MS and MS/MS data for peptide fragment identification were obtained with a MALDI-TOF/TOF instrument (4800 MALDI TOF/TOF™, Applied Biosystems, Waltham, MA, USA) and instrument parameters were set using 4000 Series Explorer™ software. MS spectra were recorded in positive reflector mode in a mass range from 500 to 4000 Da and laser intensity of 4500–4800 Hz. The spectra were calibrated using the calibration mixture mentioned above to a mass tolerance of 0.1 Da. MS/MS spectra were recorded with high-energy collision-induced dissociation and the laser intensity was increased by 10%. Precursor mass m/z 1570.65 (Glu1-Fibrinopeptide from the calibration mixture) was used to calibrate the MS/MS spectra to a mass tolerance of 0.1 Da.

#### 4.4.4. Mass Spectra Data Analysis

MS and MS/MS spectra were processed using DataExplorer™ software and the obtained output files were uploaded in the MS-PEP data analyzer (https://bioit.shinyapps.io/PARs/ accessed 10/08/2020) for peptide fragment identification. Identified fragments were confirmed using the MS/MS spectra and mMass 5.5.0 by Martin Strohalm [95]. The relative intensities of the fragments were calculated as the ratio of the intensity of the peak of interest to the intensity of the peak of the standard peptide. Confirmed fragments together with their relative intensities were used to obtain a peptide cleavage pattern and the abundance of the fragments in the reaction mixtures was compared.

### 4.5. Statistics and Figures

Statistics were performed using IBM SPSS Statistics 27 and figures were compiled in GraphPad Prism 9.2.0. To compare the abundance of peptides or formed fragments between the control, post-colitis and acute colitis groups, a Kruskal–Wallis test with a post hoc Dunnett–Bonferroni test was conducted. A Wilcoxon signed-rank test (two incubation periods) or a Friedman test followed by a Wilcoxon signed-rank test with Bonferroni correction (three incubation periods) were conducted to determine the degradation and formation of peptides and fragments in function of the incubation period. The presence of correlations between the abundance of peptide fragments after incubation with colonic tissue and measured proteolytic activities with fluorogenic substrates was evaluated with a Spearman correlation test. Results were considered significant if the *p*-value was lower than 0.05. Data are presented as mean values ± SEM.

## Figures and Tables

**Figure 1 ijms-22-10711-f001:**
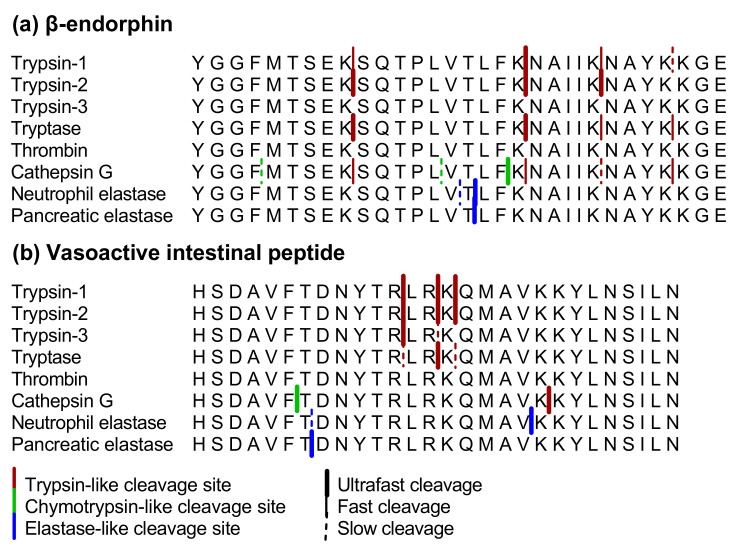
Cleavage sites in (**a**) β-endorphin and (**b**) VIP after incubation with trypsin-1, trypsin-2, trypsin-3, tryptase, thrombin, cathepsin G, neutrophil elastase or pancreatic elastase. Trypsin-like cleavage sites are shown in red, chymotrypsin-like cleavage sites are in green and elastase-like cleavage sites are in blue. Thick lines indicate ultrafast cleavages (within two min of incubation), thin lines indicate fast cleavages (after 5 min of incubation) and dotted lines indicate slow cleavages (after 10 or 30 min of incubation).

**Figure 2 ijms-22-10711-f002:**
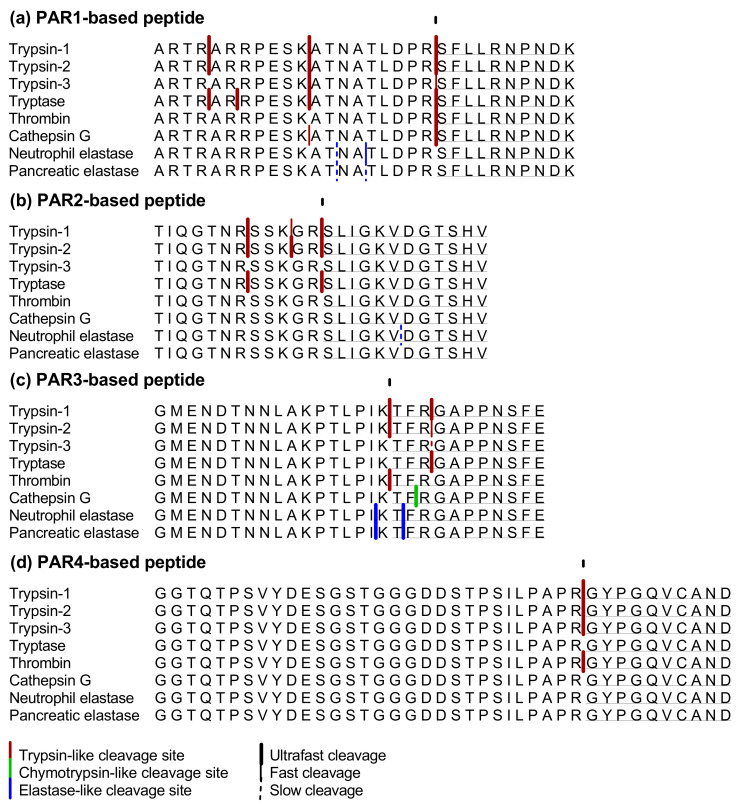
Cleavage sites in the (**a**) PAR1-, (**b**) PAR2-, (**c**)PAR3- and (**d**) PAR4-based peptides after incubation with trypsin-1, trypsin-2, trypsin-3, tryptase, thrombin, cathepsin G, neutrophil elastase or pancreatic elastase. Trypsin-like cleavage sites are shown in red, chymotrypsin-like cleavage sites in green and elastase-like cleavage sites in blue. Thick lines indicate ultrafast cleavages (within two min of incubation), thin lines indicate fast cleavages (after 5 min of incubation) and dotted lines indicate slow cleavages (after 10 or 30 min of incubation). The arrow indicates the cleavage site for the unmasking of the TL-sequence (underlined). The cleavages in the TL-sequence cause disarming of the PAR-based peptides.

**Figure 3 ijms-22-10711-f003:**
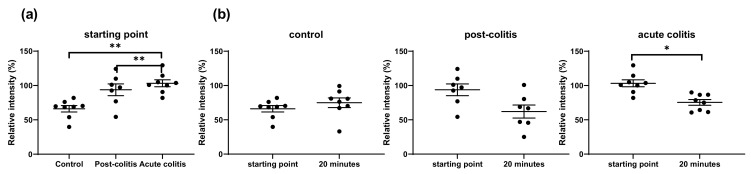
(**a**) Relative intensity of the full-length neurotensin at the starting point, after seconds of incubation with samples from control, post-colitis and acute colitis rats. Statistics: Kruskal–Wallis test post hoc Dunnett–Bonferroni. (**b**) Relative intensity of the full-length neurotensin at the starting point and after 20 min of incubation with control, post-colitis and acute colitis samples. The decrease of the full-length peptide is significant in acute colitis samples but not in control or post-colitis samples. Statistics: Wilcoxon signed-rank test. The mean and standard error of the mean (SEM) are shown. n = 8 animals per group * *p* < 0.05 *** p* < 0.01.

**Figure 4 ijms-22-10711-f004:**
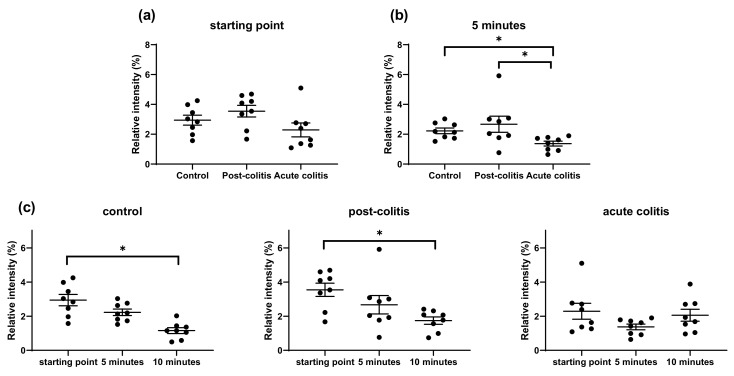
Relative intensities of the full-length Leu-enkephalin (**a**) At the starting point: There is no significant difference, but there is a trend towards lower Leu-enkephalin in acute colitis samples compared to control (*p* = 0.1) and post-colitis samples (*p* = 0.06). Statistics: Kruskal–Wallis test post hoc Dunnett–Bonferroni. (**b**) After 5 min of incubation with samples from control, post-colitis and acute colitis rats. There is a significantly lower abundance of the full-length Leu-enkephalin in acute colitis samples compared to control or post-colitis samples. Statistics: Kruskal–Wallis test post hoc Dunnett–Bonferroni. (**c**) At the starting point and after 5 and 10 min of incubation with control, post-colitis or acute colitis samples. The decrease of the full-length peptide is significant in control and post-colitis samples. Statistics: Friedman test followed by Wilcoxon signed-rank test with Bonferroni correction. The mean and SEM are shown. n = 8 animals per group * *p* < 0.05.

**Figure 5 ijms-22-10711-f005:**
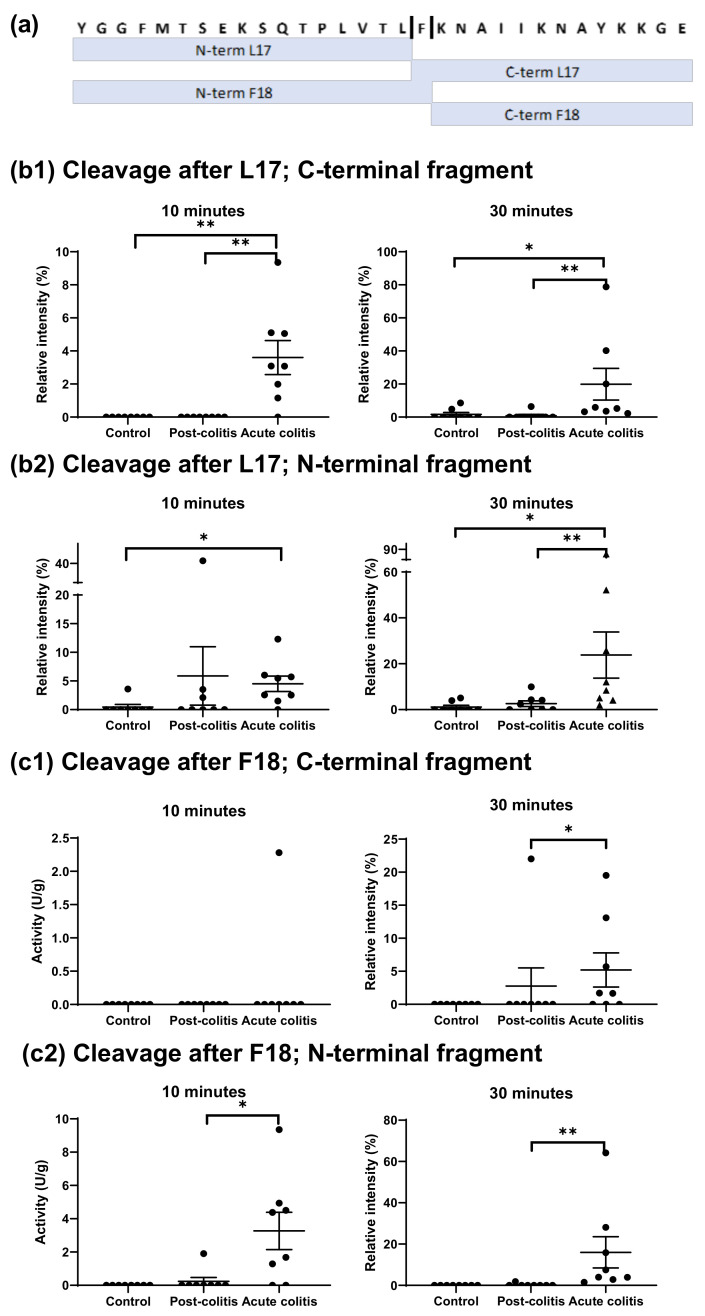
(**a**) Cleavage sites in β-endorphin after incubation with samples from control, post-colitis and acute colitis rats. (**b1**,**b2**) Relative intensities of the C- (**b1**) and N-terminal (**b2**) fragment after cleavage at leucine in position 17 after 10 and 30 min of incubation with the colonic samples. The abundance of both fragments is increased after incubation with acute colitis samples compared to control and post-colitis samples. (**c1**,**c2**) Relative intensities of the C- (**c1**) and N-terminal (**c2**) fragment after cleavage at phenylalanine in position 18 after 10 and 30 min of incubation with the colonic samples. The abundance of the C-terminal fragment is significantly higher in acute colitis samples compared to controls and the N-terminal fragment is increased in acute colitis samples compared to controls and post-colitis samples. Statistics: Kruskal–Wallis test post hoc Dunnett–Bonferroni. The mean and SEM are shown. n = 8 animals per group * *p* < 0.05 ** *p* < 0.01.

**Figure 6 ijms-22-10711-f006:**
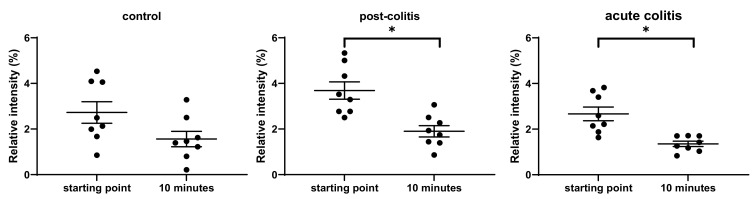
Relative intensities of the full-length Met-enkephalin at the starting point and after 10 min of incubation with colonic samples from control, post-colitis and acute colitis rats. The decrease of the peptide is significant in post-colitis and acute colitis samples but not in controls. Statistics: Wilcoxon signed-rank test. The mean and SEM are shown. n = 8 animals per group * *p* < 0.05.

**Figure 7 ijms-22-10711-f007:**
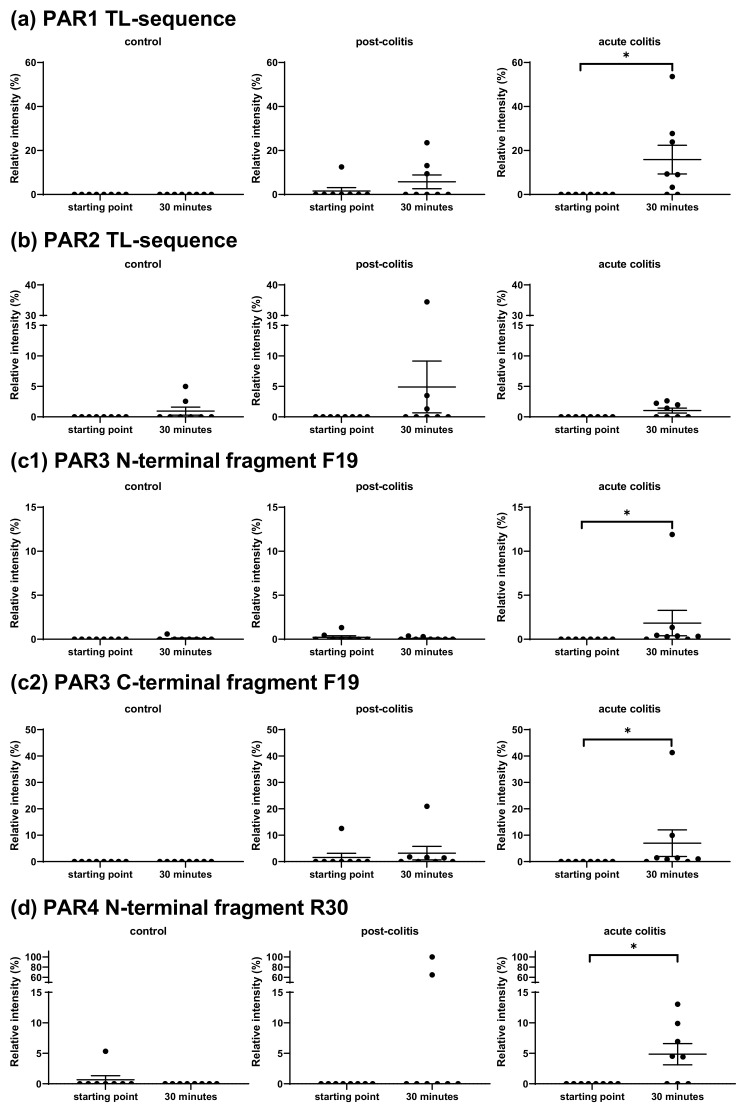
Relative intensities of (**a**) the fragment representing the TL-sequence of PAR1 after incubation of a PAR1-based peptide with colonic samples from control, post-colitis and acute colitis rats. There is a significant increase of this fragment in acute colitis samples after 30 min of incubation; (**b**) the fragment representing the TL-sequence of PAR2 after 10 and 30 min of incubation of a PAR2-based peptide with colonic samples from control, post-colitis and acute colitis rats. There is no significant difference in the abundance of this fragment between the groups (**c1**) N-terminal and (**c2**) C-terminal fragment after the disarming cleavage of the PAR3-based peptide at phenylalanine on position 19 at the starting point and after 30 min of incubation with colonic samples from control, post-colitis and acute colitis rats. The increase of both fragments is significant after incubation with acute colitis samples; (**d**) the N-terminal fragment after cleavage of the PAR4-based peptide after arginine in position 30 after incubation with samples from control, post-colitis and acute colitis rats. This fragment is significantly increased after incubation with acute colitis samples. The C-terminal fragment of this cleavage is not detected. Statistics: Wilcoxon signed-rank test. The mean and SEM are shown. n = 8 animals per group * *p* < 0.05.

**Figure 8 ijms-22-10711-f008:**
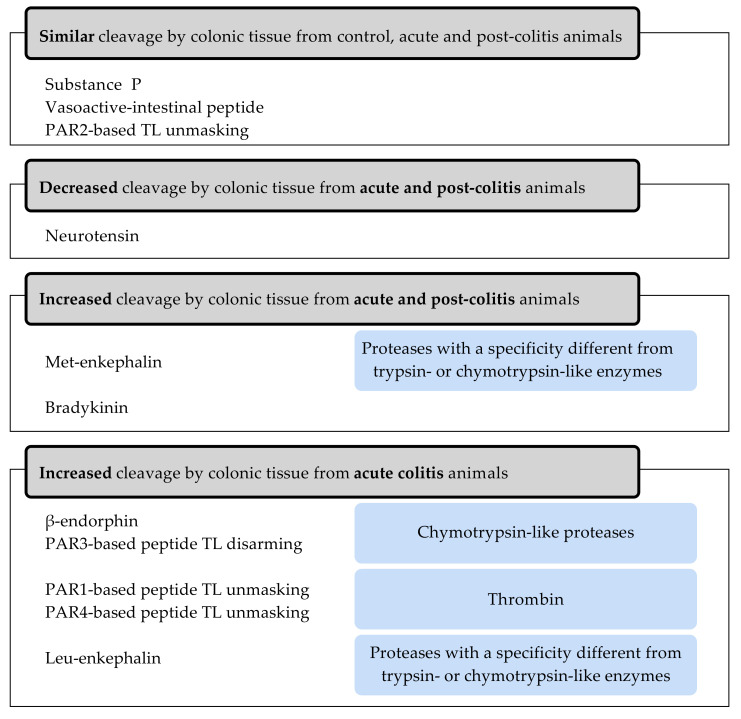
Summary of the effect of colonic tissue lysates from a TNBS-induced colitis model in the acute or post-inflammatory phase on the cleavage of bioactive peptides. The blue boxes show the hypotheses regarding proteases involved in acute or post-colitis resulting from the observed differences in peptide cleavage efficiencies by colonic tissue between the acute or post-colitis rat model and controls.

**Table 1 ijms-22-10711-t001:** Efficiency of a set of proteases (trypsin-1, trypsin-2, trypsin-3, tryptase, thrombin, cathepsin G, neutrophil elastase and pancreatic elastase) to cleave peptides related to pain and inflammation. VIP = Vasoactive intestinal peptide. The ability of the proteases to unmask (U) or disarm (D) the tethered ligand (TL)-sequence of the protease-activated receptor (PAR)-based peptides is shown as well. The efficiency of the first cleavage in the peptide is shown as +++ ultrafast processing within 2 min; ++ fast processing after 5 min; + processing after 10 or 30 min; − no processing, *N/A* not analyzed.

Peptide	Trypsin-1		Trypsin-2		Trypsin-3		Tryptase		Thrombin		Cathepsin G		Neutrophil šElastase		Pancreatic Elastase	
β-endorphin	*+++*		*+++*		*−*		*+++*		*−*		*+++*		*+++*		*+++*	
VIP	*+++*		*+++*		*+++*		*+++*		*−*		*+++*		*+++*		*+++*	
Substance P	*+*		*+*		*+*		*+*		*+*		*+*		*N/A*		*N/A*	
Neurotensin	*+*		*+*		*+*		*+*		*+*		*+*		*N/A*		*N/A*	
Bradykinin	*+*		*+*		*+*		*+*		*+*		*+*		*N/A*		*N/A*	
L-enkephalin	*N/A*		*N/A*		*N/A*		*N/A*		*N/A*		−		−		+	
M-enkephalin	*N/A*		*N/A*		*N/A*		*N/A*		*N/A*		−		−		+	
TL Unmasking or Disarming	U	D	U	D	U	D	U	D	U	D	U	D	U	D	U	D
PAR1-based	*+++*	*−*	*+++*	*−*	*++*	*−*	*+++*	*−*	*+++*	*−*	*+++*	*−*	*−*	*−*	*−*	*−*
PAR2-based	*+++*	*−*	*+++*	*−*	*−*	*−*	*+++*	*−*	*−*	*−*	*−*	*−*	*−*	*+*	*−*	*−*
PAR3-based	*+++*	*+++*	*+++*	*++*	*−*	*+*	*−*	*+++*	*+++*	*−*	*−*	*+++*	*−*	*+++*	*−*	*+++*
PAR4-based	*+++*	*−*	*+++*	*−*	*+++*	*−*	*−*	*−*	*+++*	*−*	*−*	*−*	*−*	−	−	*−*

**Table 2 ijms-22-10711-t002:** Sequences of the synthetic peptides used in the reaction mixtures for MALDI-TOF/TOF experiments. The number (#) of amino acids and the molecular weight are shown for each peptide.

Peptide of Interest	Sequence	# Amino Acids	Molecular Weight (Da)
β-endorphin	YGGFMTSEKSQTPLVTLFKNAIIKNAYKKGE	31	3465.03
VIP	HSDAVFTDNYTRLRKQMAVKKYLNSILN-NH_2_	28	3325.84
Substance P	RPKPQQFFGLM-NH_2_	11	1347.65
Neurotensin	pELYENKPRRPYIL	13	1672.95
Bradykinin	RPPGFSPFR	9	1060.22
Leu-enkephalin	YGGFL	5	555.63
Met-enkephalin	YGGFM	5	573.67
PAR1-based	ARTRARRPESKATNATLDPRSFLLRNPNDK-NH_2_	30	3452.0
PAR2-based	TIQGTNRSSKGRSLIGKVDGTSHV-NH_2_	24	2497.9
PAR3-based	GMENDTNNLAKPTLPIKTFRGAPPNSFE-NH_2_	28	3059.6
PAR4-based	GGTQTPSVYDESGSTGGGDDSTPSILPAPRGYPGQVCAND-NH_2_	40	3911.2

## Data Availability

The data presented in this study are available on request from the corresponding author.

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
