# Peer review of "Proteolytic Cleavage of Bioactive Peptides and Protease-Activated Receptors in Acute and Post-Colitis"

_ijms, 2021, doi:10.3390/ijms221910711_

Round 1
Reviewer 1 Report
The authors have investigated protease activity in colonic tissue from a rat model of colitis. The goals were to further characterise which proteases become upregulated and identify these as potential therapeutic targets for treatment of inflammatory bowel syndrome and inflammatory bowel disease. First, a panel of commercially sourced purified proteases were tested against a panel of synthetic peptide substrates and cleavage products monitored by mass spectroscopy. Then, colonic tissue extracts from control, acute colitis, and post-colitis rats were tested against the same substrates. The cleavage patterns were used to infer which proteases are upregulated in the disease model tissue and which protease-substrate interactions might contribute to pain and inflammation in the disease state.
Author Response
We thank the reviewer for the time spent on the review of our manuscript and the positive evaluation.
Reviewer 2 Report
This is an interesting study on a highly relevant topic. A few questions and suggestions:
- How did the authors choose the appropriate concentrations of enzymes and colonic lysates for use in their cleavage studies? Were these always performed at 37C? If not, then why?
- The effects of induced colitis and various enzymes on cleavage of bioactive peptide have been conclusively demonstrated. However, the corresponding results from PARs are less clear-cut.
- While PAR1 studies have shown clear results, data from the other receptors (esp., PAR3 and PAR4) are equivocal at best.
- It is suggested to use an alternative measure for PAR (2, 3, 4) studies to further support these initial conclusions. For example, cell lines transfected with different PARs could be used to demonstrate activation of downstream pro-inflammatory signaling pathway/s, by colonic lysates from control/active colitis/post-colitis samples. Without having a functional read-out in a more biologically relevant system, it is hard to appreciate the significance of these initial findings
Round 2
Reviewer 2 Report
The authors' responses to this reviewer's concerns are satisfactory.